# Cervical Twin Heterotopic Pregnancy: Overview of Ectopic Pregnancies and Scanning Detection Algorithm

**DOI:** 10.3390/medicina57090969

**Published:** 2021-09-15

**Authors:** Antonios Koutras, Zacharias Fasoulakis, Michail Diakosavvas, Athanasios Syllaios, Athanasios Pagkalos, Athina A. Samara, Georgios Tsatsaris, Thomas Ntounis, Marianna Theodora, Michael Sindos, Emmanuel N. Kontomanolis

**Affiliations:** 11st Department of Obstetrics and Gynecology, General Hospital of Athens ‘ALEXANDRA’, National and Kapodistrian University of Athens, Lourou and Vasilissis Sofias Ave., 11528 Athens, Greece; antoniskoy@yahoo.gr (A.K.); hzaxos@gmail.com (Z.F.); mdiakosavvas@gmail.com (M.D.); thomasntounis@gmail.com (T.N.); martheodr@gmail.com (M.T.); sindosgyn@hotmail.com (M.S.); 21st Department of Surgery, Laikon General Hospital, National and Kapodistrian University of Athens, Agiou Thoma Str. 17, 11527 Athens, Greece; nh_reas@hotmail.com; 3Department of Obstetrics and Gynecology, General Hospital of Xanthi, Neapoli, 67100 Xanthi, Greece; sakispagkalos@gmail.com; 4Department of Surgery, University Hospital of Larissa, Mezourlo, 41110 Larissa, Greece; 5Department of Obstetrics and Gynecology, Democritus University of Thrace, Vasilissis Sofias Str. 12, 67100 Alexandroupolis, Greece; tsatsarisg3@gmail.com (G.T.); mek-2@otenet.gr (E.N.K.)

**Keywords:** heterotopic pregnancy, cervical twins, gestation

## Abstract

*Background*: Ectopic pregnancy is the leading cause of gestation-related deaths during the first trimester. Cervical twin heterotopic pregnancies, when ectopic, constitute a small and rare part of gynecological surgery. *Case Presentation*: A 30-year-old pregnant woman (gravida 3, para 2) presented with mild pain in the lower abdomen and traces of bleeding per vaginum for three days. Transvaginal ultrasonography revealed a balloon-shaped cervical canal with a visible gestational sac measuring 3.5 × 3.9 cm. A second gestational sac was seen in the uterine cavity. The measurements of the gestational sacs corresponded to 7 + 4 weeks’ pregnancy. A decision for medical abortion with mifepristone and misoprostol was made. However, due to an incomplete abortion and continuous bleeding, a curettage was performed. *Conclusions*: Spontaneous heterotopic pregnancy with the ectopic pregnancy located in the cervix is an extremely rare clinical condition requiring urgent treatment in order to reduce maternal mortality and morbidity and preserve fertility.

## 1. Introduction

Spontaneous heterotopic pregnancy is a highly uncommon condition in which an intrauterine and extrauterine gestation occur simultaneously [1]. Cervical ectopic pregnancy (CEP) is an extremely rare condition, representing less than 0.1% of all ectopic pregnancies [2]. CEP is associated with high mother and embryo mortality and advanced pregnancies may require urgent surgical intervention, and in some cases, even a hysterectomy [3]. Diagnosis and management can be quite challenging, however, recent improvements in ultrasound resolution and earlier detection of these pregnancies have limited morbidity and preserved fertility [2,4]. As the treatment of choice (medical or surgical) for CEP remains controversial, factors such as gestational age, serum β-hCG (*Beta* Human Chorionic Gonadotropin) levels, fetal cardiac activity presence, and fertility preservation play a major role in individualization of each case [4].

Herein, we present a rare case of a spontaneous unruptured cervical twin heterotopic pregnancy treated medically in order to preserve fertility.

## 2. Case Presentation

A 30-year-old pregnant Caucasian female (gravida 3, para 2) presented with mild pain located in the lower abdomen and traces of bleeding per vaginum for three days. During the gynecological examination, blood was found in the outer cervical os and the vagina. Conception was achieved spontaneously.

Her previous menstrual cycle was normal and her gynecological history was unremarkable. She had no prior abdominal operations, and her past medical/surgical history revealed no allergies or co-morbidities; her blood group was Rhesus B (+) positive. Her obstetric history included two single pregnancies that were both delivered by a lower segment uterine section five and three years ago. Her infection profile was negative.

Ultrasonography examination was conducted thoroughly both transabdominally and transvaginally. The examination revealed a visible gestational sac measuring 3.5 × 3.9 cm and a yolk sac with a fetal pole. A second gestational sac with a yolk sac and fetal pole was seen in the uterine cavity surrounded by a thickened endometrium as expected. The measurements of both fetal poles corresponded to 7 + 4 weeks’ pregnancy. Positive cardiac activities were confirmed with the vaginal probe. In addition, magnetic resonance imaging (MRI) was requested to exclude uterine scar implantation.

Following this, a decision was made for medical abortion with 200 mg of mifepristone (mifegyne), a synthetic steroid with anti-progesteronic action, and the day after, four tablets (800 mcg) of misoprostol (cytotec), an equivalent of the prostaglandin E1, which were administered orally. Heavy bleeding was observed by the patient and she reported immediately to the emergency room (ER). An ultrasound demonstrated incomplete expulsion of the products of conception.

Curettage was performed to complete the abortion and control the bleeding originating from the network formed by the ascending branches of the vaginal artery and descending branches of the uterine artery. The bleeding was controlled with the use of a Foley balloon catheter in the cervical canal and simultaneous vaginal ligation of the branches of the cervical arteries in the vicinity of the external layer of the cervix.

## 3. Discussion

Ectopic pregnancies make up a small part of gynecological surgery, but there is an increasing incidence worldwide, causing significant maternal morbidity and mortality, accompanied by pregnancy loss [5,6,7]. In the United Kingdom, there are around 11,000 cases reported annually (11.5 per 1000 pregnancies), with a mortality rate of 0.4 per 1000 ectopic pregnancies [7]. A ruptured ectopic pregnancy is a life-threatening medical condition that requires urgent surgical intervention [8]. Spontaneous extrauterine pregnancies are considered rare, and predisposing factors include pelvic inflammatory disease (often due to chlamydia trachomatis), tobacco smoking, prior tubal surgery, history of infertility, and increased age [9]. Most ectopic pregnancies are detected in the fallopian tubes (90%) and less than 1% are located in the cervix [1,5].

Clinical suspicion of extrauterine pregnancy is raised when the patient has a positive pregnancy test, bleeding, an adnexal mass, and a conglomeration of complex fluid (suggestive of blood) in the cul-de-sac anatomical area. Free fluid is a non-specific finding, although a large amount of fluid is suggestive of an ectopic pregnancy. Complex fluid is compatible with hemoperitoneum; this is associated with ectopic pregnancy but does not necessarily signify tubal rupture [10].

Early detection of an ectopic pregnancy is vital, and laboratory investigation has limited use. When an ectopic pregnancy ruptures, hemoperitoneum, a result of the rapid accumulation of blood under pressure within the peritoneum, is an emergency situation that can lead to hemodynamic instability and that requires urgent surgery [11]. Sonographic detection and “final” diagnosis of an ectopic pregnancy are definitely the most difficult steps of the management process. Transvaginal ultrasonography (U/S) is a more accurate tool compared to the abdominal alternative in cases of cervical ectopic pregnancies (CEP) [12]. The ‘sliding sign’ on transvaginal U/S can help with the differential diagnosis of an aborting intrauterine pregnancy residing in the cervix [12,13,14]. The presence of this kind of pregnancy becomes noticeable with obturator pain. Yet, despite improvements in ultrasonography, the consequences of misdiagnosis (unnecessary surgery, interruption of a normal and viable gestation) are still likely to result [15]. A diagnostic laparoscopy is the gold standard for the diagnosis of an ectopic gestation [16].

The uterine scar due to a previous cesarean section is closely located to the internal os. In our case, the patient had undergone two previous cesarean sections, which partially justified the presence of the cervical pregnancy without, however, being implanted in the previous scar. To exclude uterine scar implantation, an MRI was performed.

Transvaginal U/S seems to be more sensitive in the definitive diagnosis of hemoperitoneum than culdocentesis f; we could otherwise conclude that transvaginal sonography has to a great extent replaced culdocentesis in the evaluation of ectopic pregnancy. Sonographic diagnosis of a cervical pregnancy is now established when a gestational sac surrounded by a peritrophoblastic flow is observable, or when a live embryo with positive cardiac activity is detected within the cervical canal. Moreover, the endometrium might have a pseudogestational sac with decidual cysts around it [17]. Neither type of pain nor the location of pain is specific for confirming the location of an ectopic pregnancy [18]. In our case, the patient presented to our external department with pain located in the lower abdomen and vaginal bleeding (spotting) for the past three days that she did not pay attention to, due to the fact that she had previously experienced the same symptom without any particular findings after gynecological examination.

A negative hCG essentially excludes the diagnosis of a live pregnancy, although a chronic ectopic pregnancy may be present. Normally, serum beta-hCG becomes positive at around 23 menstrual days (nine days postconception). In a normal pregnancy, the hCG doubling time is two days. In a twin pregnancy, hCG rises rapidly from the very beginning of gestation. In ectopic pregnancy, the doubling time is prolonged. If hCG levels are abnormally elevated, the patient might have an ectopic pregnancy [19].

Approximately 60% of surgically treated patients subsequently experience a term pregnancy.

Heterotopic multifetal pregnancy is the co-existence of an intrauterine pregnancy with an ectopic pregnancy. All multiple gestations should be carefully evaluated for the possibility of an ectopic pregnancy [20]. The sac should be eccentric to the endometrial cavity. The earliest sign of an intrauterine pregnancy is a small fluid collection in the endometrium [21]. Ectopic pregnancy will eventually lead to adverse outcomes and a decision for the intrauterine pregnancy has to be made. In the literature, there have been described many cases of a live birth after a successfully treated heterotopy pregnancy [1]. In our case, where the heterotopy pregnancy was located in the cervix, the patient decided to stop both pregnancies, and misoprostol was administered followed by dilation and curettage due to severe bleeding.

The cervix is composed predominantly of fibrous tissue, and patients bleed profusely. Currently, conservative treatment includes sonographically introduced local potassium chloride injection, systemic or local methotrexate, or preoperative uterine artery embolization before dilatation and curettage, with the aim to preserve future reproductive potential [22].

## 4. Conclusions

Cervical twin heterotopic pregnancies, when ectopic, constitute a small part of gynecological surgery; delayed diagnosis and treatment can lead to mothers’ mortality and embryo loss. Transvaginal scanning seems to be an accurate and safe way of detecting hemoperitoneum and gaining a better look at the endometrium, while sonographic diagnosis is not that clear in all cases. However, the most efficient method of diagnosing and managing ectopic pregnancies is still under investigation.

## Data Availability

Data are available upon reasonable request.

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
