# Peer review of "Cervical Twin Heterotopic Pregnancy: Overview of Ectopic Pregnancies and Scanning Detection Algorithm"

_medicina, 2021, doi:10.3390/medicina57090969_

Round 1

Reviewer 1 Report

The corrections that have done improved the quality of the manuscript

Author Response

We would like to thank the reviwer for the kindly stated comments.

Furthermore, our manuscript underwent an English language editing.

Reviewer 2 Report

Overall with the corrections done on this case report except for additional editing of the English.

Author Response

We would like to thank the reviwer for the kindly stated comments.

Furthermore, our manuscript underwent an English language editing.

This manuscript is a resubmission of an earlier submission. The following is a list of the peer review reports and author responses from that submission.

Round 1

Reviewer 1 Report

The article is a presentation of a case of heterotopic cervical pregnancy, with some thoughts on ectopic pregnancies in general. The introduction is very incomplete and does not refer to general data of heterotopic and cervical pregnancy with the relevant bibliographic references. The presentation of the case is very incomplete with many ambiguities in the description: The authors state that the gestational age was 8 weeks, while the patient had only 1 week of amenorrhea and also that the previous menstruation was normal. It is obvious that all 3 events cannot be valid at the same time. Also the authors do not clearly state the Obstetric History of the woman and specifically if the previous pregnancy was twin or if there were 2 single pregnancies. The discussion generally refers to ectopic pregnancy and not as it should to the strong point of the case which is the heterotopic pregnancy with cervical location.
Line 83: Please replace "Conception was achieved through the physical way", with The conception was achieved spontaneously.
Line 92: Please state the mgs of misoprostol.
I think it is better to replace everywhere the term extrauterine with the term ectopic.
Line 109: please replace - with ,
Line 132: Of course the diagnostic laparoscopy is the gold standard for the diagnosis of the ectopic pregnancy, but not for the cervical one.
Line 137: I am not aware for the meaning of the term complex fluid.
It is an interesting case in a hot topic of Obstetrics & Gynaecology, but the manuscript needs serious improvement in order to be accepted. You must focus on the heterotopic and the cervical pregnancy and in the Discussion you must focus on your findings and your management with the respective ones of the international literature.

Author Response

After thanking the reviewer for the kind and well-stated comments, we would like to answer each one separately.

  1. The article is a presentation of a case of heterotopic cervical pregnancy, with some thoughts on ectopic pregnancies in general. The introduction is very incomplete and does not refer to general data of heterotopic and cervical pregnancy with the relevant bibliographic references.

Authors’ answer: We have modified the introduction section and relevant references were added in accordance with the well-stated reviewer’s concern.

  1. The presentation of the case is very incomplete with many ambiguities in the description: The authors state that the gestational age was 8 weeks, while the patient had only 1 week of amenorrhea and also that the previous menstruation was normal. It is obvious that all 3 events cannot be valid at the same time.

Authors’ answer: We have modified the case presentation section and more details regarding gestation age were added. The pregnancy was 7 weeks and 4 days. By typo errors these ambiguities were stated. Thank you for this well-stated concern.

  1. Also the authors do not clearly state the Obstetric History of the woman and specifically if the previous pregnancy was twin or if there were 2 single pregnancies.

Authors’ answer: Details regarding previous Obstetric history (2 single pregnancies) were added.

  1. The discussion generally refers to ectopic pregnancy and not as it should to the strong point of the case which is the heterotopic pregnancy with cervical location.

Authors’ answer: Thank you for this comment. Discussion section was modified and information for cervical heterotopic pregnancies were included.

  1. Line 83: Please replace "Conception was achieved through the physical way", with The conception was achieved spontaneously.

Authors’ answer: The present phrase was modified accordingly.

  1. Line 92: Please state the mgs of misoprostol.

Authors’ answer: Dosage of misoprostol was added.

  1. I think it is better to replace everywhere the term extrauterine with the term ectopic.

Authors’ answer: The present phrase was modified accordingly.

  1. Line 109: please replace - with ,

Authors’ answer: The present phrase was modified accordingly.

  1. Line 132: Of course the diagnostic laparoscopy is the gold standard for the diagnosis of the ectopic pregnancy, but not for the cervical one.

Authors’ answer: Thank you for this comment. Diagnostic laparoscopy is the standard of treatment for ectopic pregnancies in general. This phrase was modified accordingly.

  1. Line 137: I am not aware for the meaning of the term complex fluid.

Authors’ answer: The present phrase was modified accordingly in order to avoid misunderstandings.

  1. It is an interesting case in a hot topic of Obstetrics & Gynaecology, but the manuscript needs serious improvement in order to be accepted. You must focus on the heterotopic and the cervical pregnancy and in the Discussion you must focus on your findings and your management with the respective ones of the international literature.

Authors’ answer: Thank you for the positive comments. In accordance with this well-stated comment, discussion section was modified and focused on our case and related literature.

Reviewer 2 Report

Overview: I think that this case report had great potential. A heterotopic pregnancy with a normal intrauterine pregnancy and a cervical ectopic pregnancy is a rare event. Overall your descriptions are OK, however you have missed the chance to have a significant case report. Most of your discussion is about the diagnosis and management of an ectopic pregnancy of which there are countless reports in the literature. You have a unique opportunity to report the diagnosis and management of a heterotopic pregnancy with the ectopic being in the cervix. A proper literature search with search engines identified, search terms, years of the search would help you define the rarity of this combination and treatment options. 

Specific comments: Line 75, after 30 year old caucasian I would put her gravidity,  parity, and whether she has had any abortions. Line 81 would clearly state if both of her prior pregnancies were delivered by cesarean. Line 83, may be better to say Conception was achieved naturally. line 88-89, need to state the positive cardiac activity was observed in "both" gestational sacs. line 95 instead of embryo's remmants would suggest "products of conception". Line 98 and 99 instead of Reduction of bleeding was performed would suggest "Bleeding was controlled with the use of a Foley..." LIne 104 instead of escorted by embryo loss would suggest " and pregnancy loss". Line 110 ectopic pregnancies are "in" the fallopian tube not "on" the fallopian tube. lines 112 to 119 talk about the compression of the pelvic organs, the compression on the pelvic organs is minimal, at best, the cause for concern is the bleeding from the ectopic that results in hemodynamic compromise and serious maternal morbidity or mortality if prompt surgical intervention is not undertaken. LIne 121 to 125 talk about the superiority of the vaginal vs. abdominal US. This is certainly true for a cervical ectopic but not necessarily for an ectopic in the fallopian tube or an abdominal ectopic pregnancy. Line 156; Approximately 60% of surgically treated patients ... this sentence needs to be referenced.  

Author Response

After thanking the reviewer for the kind and well-stated comments, we would like to answer each one separately.

  1. Overview: I think that this case report had great potential. A heterotopic pregnancy with a normal intrauterine pregnancy and a cervical ectopic pregnancy is a rare event. Overall your descriptions are OK, however you have missed the chance to have a significant case report. Most of your discussion is about the diagnosis and management of an ectopic pregnancy of which there are countless reports in the literature. You have a unique opportunity to report the diagnosis and management of a heterotopic pregnancy with the ectopic being in the cervix. A proper literature search with search engines identified, search terms, years of the search would help you define the rarity of this combination and treatment options.

Authors’ answer: Thank you for the positive comments. In accordance with this well-stated comment, introduction and discussion sections were modified and focused on our case of a cervix ectopic pregnancy and related literature on management of these cases.

  1. Specific comments: Line 75, after 30 year old caucasian I would put her gravidity, parity, and whether she has had any abortions.

Authors’ answer: Gravidity and parity were added accordingly to the present well-stated comment.

  1. Line 81 would clearly state if both of her prior pregnancies were delivered by cesarean.

Authors’ answer: Both of previously pregnancies were delivered by cesarean and this point was modified accordingly.

  1. Line 83, may be better to say Conception was achieved naturally.

Authors’ answer: This point was modified accordingly.

  1. line 88-89, need to state the positive cardiac activity was observed in "both" gestational sacs.

Authors’ answer: This point was modified accordingly.

  1. line 95 instead of embryo's remmants would suggest "products of conception".

Authors’ answer: This point was modified accordingly.

  1. Line 98 and 99 instead of Reduction of bleeding was performed would suggest "Bleeding was controlled with the use of a Foley..."

Authors’ answer: This point was modified accordingly.

  1. LIne 104 instead of escorted by embryo loss would suggest " and pregnancy loss".

Authors’ answer: This point was modified accordingly.

  1. Line 110 ectopic pregnancies are "in" the fallopian tube not "on" the fallopian tube.

Authors’ answer: This point was modified accordingly.

  1. lines 112 to 119 talk about the compression of the pelvic organs, the compression on the pelvic organs is minimal, at best, the cause for concern is the bleeding from the ectopic that results in hemodynamic compromise and serious maternal morbidity or mortality if prompt surgical intervention is not undertaken.

Authors’ answer: Thank you for this well-stated comment. With no doubt hemodynamic instability due to heavy bleeding is the major cause of maternal mortality and morbidity. These points were modified accordingly.

  1. LIne 121 to 125 talk about the superiority of the vaginal vs. abdominal US. This is certainly true for a cervical ectopic but not necessarily for an ectopic in the fallopian tube or an abdominal ectopic pregnancy.

Authors’ answer: This point was modified accordingly.

  1. Line 156; Approximately 60% of surgically treated patients ... this sentence needs to be referenced.

Authors’ answer: A reference was cited in this point.